# The Practice of *Rou* 柔 from Wang Bi's Perspective

## Limei Jiang

Center of Value & Culture, School of Philosophy, Beijing Normal University, Beijing 100875, China;
jianglimei@bnu.edu.cn

**Abstract:** This paper holds that Laozi's philosophy on softness is a topic that remains to be fully discussed. By distinguishing between the meanings of softness and weakness, this paper discusses how Wang Bi semantically integrated the two, presenting them as methods to attain the Dao. In this paper, the differences in Wang Bi's usage of "柔弱" (softness and weakness) and "柔顺" (softness and compliance) in his annotations on the *Daodejing* and *Yijing* are noted, emphasizing the logical support and rational explanation that Wang Bi provided for the external behaviors of gentleness described in the hexagram lines. Wang Bi reconciled the contradictions between Confucian and Daoist views on valuing gentleness and balancing Yin–Yang. In the text, he elaborates on gentleness as both a personal moral requirement and a method of social governance, addressing the real-world issues of his time and thus greatly enhancing the practicality of Laozi's philosophy of valuing softness.

**Keywords:** Yin–Yang; nothingness; non-action; virtue

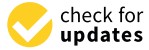



The school of Zhuangzi described that Guan Yin 关尹 and Lao Dan 老聃 had the appearance of pliant weakness and self-deprecating humility (Chapter Under the Heaven, Ziporyn 2009, p. 123), *Lūshi Chunqiu* 吕氏春秋 also summarized the philosophy of Lao Dan as *Rou* 柔 (softness)[1], and Huainanzi 淮南子 held that softness and weakness are the essence of the Dao. Later, both the *Hanshu* (汉书 History of the Han Dynasty) and the *Siku Quanshu* (四库全书 Complete Library of the Four Treasuries) took it as the main content of Taoist thought[2]. Even though Laozi's philosophy on softness is highly appreciated, the conclusion that it should be regarded as the core of Laozi's thought has not been widely accepted; therefore, we are more inclined to summarize Laozi's philosophy with concepts like Dao, non-action, naturalness (自然), etc.

In the *Daodejing* 道德经, the term "Rou" appears 11 times and in half of these instances, it was paired with Ruo 弱 (pliancy)[3] to form the term "rouruo" 柔弱 (soft and pliant). The text closely associates "柔" with the related ideas of *xu* 虚 (emptiness) and *pu* 朴 (simplicity). However, if we differentiate Laozi's thoughts into two approaches, theoretical construction and action philosophy, we find that Rou becomes an important channel that connects the two paths. By using metaphors such as infants and water, Laozi demonstrates the function of being Rou and the principle of how "the weak overcomes the strong", treating "sticking to Rou" as an important manifestation of the great Dao. Later commentaries usually regarded Rou as the idea of action and developed different modes of understanding Rou. As a representative commentator of the *Daodejing*, Wang Bi internalized Laozi's concept of Rou into his interpretation of the *Yijing*. This paper will explore the new mode that Wang Bi 王弼 applied in his commentary and his elaborations on how the Rou can be put into practice. This paper will also analyze the action philosophy that developed from Laozi's thoughts through annotations, and elucidate the practical significance of Daoist thought.

## 1. A Propositional Attitude to Be Demonstrated

In the *Daodejing*, Rou (softness) was usually related with the weak, the female, which represents a power of negativity and obedience. Laozi argued about the great power of Rou and stated that "The soft is overcoming the hard", which is quite different from conventional values and the development of the history of his period. In the Pre-Qin dynasty,

faced with Emperor Zhou's loss of control, the people in power strived to strengthen the national power with military force in order to obtain short-term benefits and achievements. Just as Sima Qian described, at that time, Qin used Shang Yang's legalist reforms to enrich the country and strengthen its military; Chu and Wei used Wu Qi's military strategies to defeat weaker enemies; King Xuan of Qi employed the disciples of Sun Tzu and Tian Ji, resulting in the vassals facing east to Qi. The world was struggling with attacks due to the opposite theories of the vertical and horizontal alliance of states. "当世之时，秦用商君，富国强兵；楚魏用吴起，战胜弱敌；齐宣王用孙子田忌之徒，而诸侯东面朝齐，天下方务于合纵连横，以攻伐为贤" (史记 *Records of the Grand Historian*). According to Sima, worshipping the strong and the powerful was the conventional value among most politicians and intellectuals. Various schools of thought such as military strategists, legalists, Mohists, etc., put forward similar methods in response to the demands of reality. The struggle for hegemony in the Pre-Qin period strengthened the role of power in enhancing national strength.

Compared with these ideas, discourses to support the female and the weak held by Laozi seem to be implausible. Chad Hansen called it "a heuristic corrective to our value preference" (Hansen 1992, p. 225). The principle of softness cannot be verified by the development of social reality, and softness is usually associated with "weakness", which means it represents concepts of being conquered and defeated. From the few rare expositions on softness in the *Daodejing*, Laozi's discussion on "softness" resembles a propositional attitude more than a causal argument. He tried to explain the existence of the utmost softness by using metaphors like infants and water. The initial imagery of the character "柔" (softness) originates from "wood", but the imagery of wood is clearly definitive which cannot fully express the flexibility that Laozi wanted to convey with "Rou". Therefore, through corrective thinking, Laozi clarified that gentleness is an inherent attribute of every entity:

> Focus your vital breath until it is supremely soft, can you be like a baby? (Chap. 10, Mair 1990, p. 181)

> Nothing in All Under Heaven is more supple and soft than water. (Chap. 78, Mair 1990, p. 153)

Although an infant's muscles and bones are soft and delicate, they can grip firmly without intending to, retaining the abundant energy of life. When alive, a person is soft and flexible, but becomes rigid after death. Life is like flowing water, continuous and unending, flowing ceaselessly, using its never-ending strength to overcome seemingly hard obstacles. Just like water flowing around obstacles, a soft and yielding approach can be more effective than a forceful one.

> The soft and weak conquer the strong. (Chap. 36, Mair 1990, p. 260)

Through these statements, Laozi came to the conclusion that "softness overcomes hardness", which is derived from his observations and insights into nature. In the contrast between hardness and softness, he placed greater emphasis on the power of "softness". However, in Chapter 2, Laozi stated that opposing terms, like beauty/ugliness, goodness/badness, long/short, and high/low, can change into each other. Accordingly, in the pair of hardness and softness, softness cannot obtain a position higher than hardness. It seems that the rule of "stick with Rou" contradicts the principle of "negative opposite (反 fan)" (Wagner 2003, p. 257) in the *Daodejing*. According to the *Shuowen Jiezi* 说文解字, Rou describes two different statuses of wood or trees: bending and straightening, which are derived from the direct human experience of trees (Wang 2022, p. 202). It implies that the meaning of Rou itself has attributes of both softness and hardness. Duan Yucai 段玉裁 further pointed out that wood that is bent can be straightened, and that which is straight can be bent; this is called "softness". Softness precisely refers to the characteristic of trees and wood being flexibly adaptive between being bent and straight. By using the word "overcome", Laozi demonstrated the rule that the softest is also the strongest. Within Laozi's philosophy, the idea of maintaining softness also has certain contradictions with his theory of *wuwei* (non-action). The concept of *wuwei* stresses following the natural attributes of things, not intervening artificially in their development. This means that, if something's in-

trinsic nature is rigid and unbending, then we should not expect it to live in a soft manner. However, from another perspective, as understood by Eric Santner (2001, p. 21), softness also signifies a life that is "a kind of withdrawal from", reconstructing unconventional social adaptation and social bonds by forgoing confrontation. Chen Guying 陈鼓应 further pointed out that "softness and weakness describe how the Dao operates without a sense of pressure" (Chen 1984, p. 226). Laozi's arguments on softness have a unique historical perspective and practical significance. He evolved the early Daoist self-centered philosophy of avoiding harms throughout one's life into an art and skill of living, as well as an essential method for political interactions and conflicts. As recorded in the chapter *Yang Zhu* of *Liezi* 列子, "The ancients would neither harm the world for a slight personal gain nor sacrifice their personal well-being for the world. If everyone refrains from harming others and from exploiting the world, then the world will be in order". Yang Zhu 杨朱 advocated an extreme form of self-interest as his target, prioritizing one's own welfare over everything else. Laozi's philosophy on softness, however, suggested avoiding harm and disturbances from the external world as much as possible by adopting a flexible approach. Compared with Yang Zhu's approach, this is much more reasonable from the perspective of self-preservation. Laozi established a new type of relationship between the self and others through the method of softness. Softness does not mean letting oneself become weak, subordinate, or overlooked. On the contrary, as an intrinsic quality of the self, compared to the confrontational characteristics of "hardness", "softness" possesses greater endurance and resilience due to its accommodating nature. It fosters more compassion in relationships with others and offers a "negative opposite (反)" perspective, allowing everything to be examined from the viewpoint of the differences between parties. Therefore, in terms of the wholeness of things, such a self experiences a more complete existence and, in another sense, is stronger, with the potential to overcome the mighty. From a political perspective, Laozi did not believe there was a fundamental difference between the prevalent values of strongness and hardness versus those of gentleness and softness. For a small country to survive amidst the chaos of warring states, it must first adopt a gentle approach to diplomacy to preserve itself. Meanwhile, for a large country to conquer a smaller one, it must "position itself beneath the smaller country", handling relations with other nations with a humble attitude.

Laozi did not provide a comprehensive discourse supporting his advocacy of valuing softness. His limited discussions largely originated from observations of life experiences, considering softness as an inherent principle of Dao. This has necessitated subsequent followers, in their commentaries, to provide thorough explanations for the reasons and rationale for "softness", and to offer detailed plans for its practical application. Before the commentary of Wang Bi, we can see that authors had already explored some explanatory modes. As to the philosophy of softness, Zhuangzi fundamentally accepted Laozi's views. He regarded the art of softness as a manifestation of the Dao and elaborated on its significance with a plethora of stories, highlighting the dangers of rigid dominance in governing a country and in personal well-being. However, he did not present much theoretical explanation on this. On the other hand, Han Fei 韩非, a representative of the legalist school of thought, pointed out that the Dao is also weak and gentle, because it is the general basis for the development of all things. In the chapter *Interpretations of Dao* (解老 *Jie Lao*), he said "The state of Dao, does not control or manifest, soft and weak at any time, correspond with the principle. 凡道之情，不制不形，柔弱随时，与理相应". He believes that the Dao does not necessarily always manifest as a display of strength. An individual can, based on specific circumstances, adopt a gentle and vulnerable posture to handle the current situation, and this precisely aligns with the demands of the Dao. In the chapter *Metaphoric Interpretations of Laozi* (喻老 *Yu Lao*), he explained the idea of "keeping soft is stronger" 守柔曰强 with the historical stories of Gou Jian's 勾践 washing horses for King Wu and King Wen being sworn at the gate of King Zhou[4]. Han Fei regarded softness as an effective way to achieve political success, highlighting its connotation of enduring humiliation. Moreover,



the "softness" with strategic nuances that he developed actually reinforced the aspect of Laozi's art of governing by doing nothing.

In the Han Dynasty, Yan Zun 严遵 explained the significance of "softness and weakness" in life based on the theory of *Qi* (气 a vital force) that prevailed during the Han dynasty. He said:

> Where Yang Qi 阳气 resides, wood can be alleyways and grass can be formed. Where Yang Qi departs, water can be condensed and ice can be broken. Therefore, the divine and Yang Qi is the root of life. Softness and weakness are the medicine for all things. Gentleness and vulnerability are the tools for growth, and they attract the divine and yang energy. All things follow the yang energy with softness and weakness. Thus, rigidity and strength are the manifestations of death, while gentleness, weakness, and smoothness are the disciples of life.

> 阳气之所居，木可巷而草可结也。阳气之所去，水可凝而冰可折也。故神明阳气生之根也。柔弱物之药也。柔弱和顺生长之具，而神明阳气之所托也。万物随阳气以柔弱也，故坚强实死之形象，柔弱润滑生之徒类也

Yan Zun emphasized the importance of *Yang Qi*，and treated softness as a specific attribute of Yang Qi. In the cycle of life and nature, while too much rigidity or strength can lead to stagnation or decay, it is "softness and weakness that revitalizes everything, acting like medicine". Compared to Han Fei, Yan Zun's discourse treats softness as a philosophy for nurturing life. He emphasized its significance in sustaining life energy and does not overly extend its implications into the realm of politics. Later, in the comments of Heshang Gong 河上公, "Softness and Submissiveness" 柔顺, "Softness and Gentleness" 柔和 and "gentleness" were often used to describe "softness". With these descriptive words, he highlighted the softness of the body and interpreted "softness" as a virtue of gentleness and humility. Hardness and softness, as a pair, derive from people's empirical cognition of the world. They summarize two different concepts from opposite phenomena in the world. As He Shanggong 河上公 said, "The world is being named for its physical position, yin and yang, and softness and hardness". 天地有形位、有阴阳、有柔刚，是其有名也。 (Commentary to Chapter 1). Therefore, Yin–Yang, together with hardness–softness, provide people with different perspectives for observing the world, but in the early period, people did not seek the consistency between the two, nor did they replace the latter with the former.

Laozi's concept of softness has also been questioned by later generations. For instance, Xunzi 荀子 criticized him, saying, "Laozi saw the value of yielding, but not the value of exerting oneself… If there is only yielding and no exerting oneself, noble and lowly will not be distinguished. 老子有见于诎，无见于信……有诎而无信，则贵贱不分" (Hutton 2014, pp. 181–82). Here, the term "qu" (诎) originates from bending, and "xin" (信) corresponds with 'stretch.' Thus, qu and xin are synonyms for bending and stretching, metaphorically referring to weakness and strength. In the *Bu Gou* (不苟 Nothing Improper) chapter, Xunzi pointed out the principle of "bending and straightens with the occasion" (Hutton 2014, p. 17), suggesting that as a gentleman, one should be broad-minded without being lax, upright without being rigid, articulate without being argumentative, observant without being provocative, standing firm without being oppressive, strong without being violent, adaptable without losing direction, and respectful and cautious while being accommodating. He was an advocate for the spirit of "softness without yielding". In other words, although he recognized the value of softness, he advocated more for the value of firmness and resilience, believing in the principle that "firmness and tenacity can be trusted in all circumstances", encouraging people to uphold the Confucian aspiration to save the world even in difficult situations. According to Xunzi, the issue with Laozi's philosophy of softness lay in its lack of spirit in upholding greater principles, making it unable to guide people in making decisions of yielding or exerting in specific situations. Laozi undermined conventional assumptions by reversing the priorities in the standard oppositions (Graham 1986, p. 5). He regarded "weakness" as the external manifestation

of using Dao in Chapter 40. In Chapter 78, he also pointed out the problem that it is easy to know softness but difficult to put it into practice. Theoretical models of Qi and Yin–Yang in the commentaries only provided a metaphysical argument for the theory of softness. However, the results of these annotations only served to emphasize the power of softness. They did not provide detailed guidance on how to understand the function and how to apply the methods of softness.

## 2. Softness and Pliancy (柔弱) vs. Softness and Compliance (柔顺)

Although "柔" (soft) and "弱" (pliant) are often used together in the *Daodejing*, there is a distinction between them. Based on our previous explanation, softness refers to the ability of trees and wood to transition between being bent and straight. As Duan Yucai explained, "Pliancy means to bend. A bending object refers to bent wood, and is extended to denote anything that bends". The upper part of the character "弱" (weakness) with "弓" represents the appearance of bending; while the "彡" in it symbolizes tufts or flowing hair. Duan holds that weak things 弓 cannot stand alone, that is why they must be connected in pairs. In a word, "弱" (pliancy) only represents one state of soft wood while "柔" refers to two. However, Wang Bi placed a greater emphasis on the phonetic similarity of the two rather than the difference between the meanings. In trying to provide a linguistic explanation of the consistency between softness (柔 Rou) and pliancy (弱 Ruo), he said:

> Being soft and weak penetrates [the other entities] in likewise manner, without [oneself] being exhaustible. (Wagner 2003, p. 257) 柔弱相通，不可穷极。

"Rou" and "Ruo" have quite similar pronunciations in ancient China. Gu Yanwu 顾炎武 also stated in *Tang Dynasty Rhyme Regulation* (唐韵正 *TangYun Zheng*) that Rou 柔 was pronounced as Ru 蠕. Emperor of Wei (拓跋嗣 Tuoba Si) changed the pronunciation of "Rou Ran" (柔然) into "Ru Ru" (蠕蠕). According to the historical records, "Rou Ran" (柔然) was called "Ru Ru" (蠕蠕) in the Northern Wei Dynasty, and later people in the South called it "Rui Rui" 芮芮. In the text of Wang Bi's commentary on the Laozi, the term "柔弱" (soft and pliant) appears four times, but we did not find "柔顺" (gentle and compliant) even once, which demonstrates that Wang Bi tended to regard softness as being synonymous with pliancy.

On the basis of semantic analysis, Wang Bi advocated Laozi's opinion that the soft and the pliant provide life with continual and inexhaustible energy, possessing the power to overcome the rigid and strong, and provided a theoretical explanation for this. In his commentary on Chapter 43 of Laozi, he said:

> For ether there is nothing that it does not penetrate; for water there is nothing that it does not get through. There is nothing that the empty and negative as well as the soft and weak do not penetrate. That which is without entity is inexhaustible; the softest cannot be broken. He is extrapolating from these [two]; that is why [the Laozi says] "surmise" that "non-interference brings benefits" for [the other entities]! (Wagner 2003, pp. 268–69) 气无所不入，水无所不经。虚无柔弱，无所不通。无有不可穷，至柔不可折。以此推之，故知无为之有益也。

Wang Bi believed that the reason why softness and weakness possess such great power is due to their characteristics of non-being (无 wu), similar to water and air, which are formless and boundless. In the commentary on Chapter 55 of Laozi, he said:

> An infant is without cravings and without desires, and [thus] does not offend the multitude of [other] beings. That is why it is a human being which [in turn] beings like poisonous insects will not offend. [The ruler] who has the fullness of capacity in himself will not offend the multitude of other entities. That is why there will be no other entity detrimental to his intactness. (Wagner 2003, p. 305)

> 赤子，无求无欲，不犯众物，故毒螫之物无犯于人也。含德之厚者，不犯于物，故无物以损其全也。

An infant is, physiologically speaking, an extremely delicate entity. However, Laozi compared virtuous individuals to infants and believed that those with virtue possess a mysterious power in their individual lives. They are impervious to harm from venomous snakes, fierce beasts, and birds. Wang Bi pointed out that Laozi described virtuous individuals as infants because infants are free from desires and demands (much as when we refer to people today as having a "childlike heart"). Since they lack purpose in their actions, they naturally do not interfere with the unfolding of events, which allows them to harmoniously coexist with all things and preserve themselves. According to Wang Bi, the Dao is the very reason and basis for the existence of all things, and things can fully present themselves only when they return to non-being.

The intrinsic nature of "nothingness" manifests in specific utilities, exhibiting characteristics such as namelessness, absence of desire, non-action, and formlessness. They no longer become a particular, biased existence and thus can display the characteristics of the Dao. Wang Bi's discussion of "softness" was based on a cosmic perspective that emphasizes "non-being". He viewed "softness" as a crucial component of the power of "nothingness". This makes the pursuit of the Dao equivalent to the practice of softness. Such a model essentially elevates the rationale underpinning the act of being soft. The model also breaks through doubts generated in binary oppositions by adhering to one side (softness), making it a conscious pursuit for humans.

Wang Bi also replaced Laozi's empirical summaries with logical reasoning to explain the principle that the soft can overcome the hard. In the *General Remarks on the Changes of the Zhou* (周易略例 *Zhouyi Lueli*), in order to explain the changes in the hexagrams (爻), Wang Bi pointed out that there exists something universal in things, namely the contradiction between the material itself and its inclination. He said:

> What is change? It is what is brought about by the interaction of the innate tendency of things and their countertendencies to spuriousness. The actions of this tendency to spuriousness are not to be sought in numbers [i.e., they are beyond count]. Thus when something that tends to coalescence would disperse or when something that tends to contraction would expand, this runs counter to the true substances involved. In form a thing might seem inclined to agitation yet wants to be still, or a material though soft still craves to be hard. Here, substance and its innate tendency are in opposition, and material and its inclination are in contradiction. ([Lynn 1994](#), p. 27)

> 变者何也？情伪之所为也。夫情伪之动，非数之所求也; 故合散屈伸，与体相乖。形躁好静，质柔爱刚，体与情反，质与愿违。

There is a tension between the actual state of existence of things and their inherent nature, which pushes things to develop in the opposite direction. According to *The Commentary on the Appended Phrases* (系辞 Xici zhuan) Part Two, "It is when true innate tendencies and spurious countertendencies work their influence that advantage and harm are produced. {If things respond to true innate tendencies, 'advantage' will obtain, but if things respond to spurious counter-tendencies, 'harm' will prevail.}" ([Lynn 1994](#), p. 95). Thus, things that seem to be growing and developing continuously should not only be admired for their robust virtues but should also pursue the path of softness and pliancy. Wang Bi's discussion on how the soft overcomes the hard takes the change and development of things as its premise. This emphasis on softness in change aligns with the Yijing's emphasis on change, making it possible for Wang Bi to create a bridge between the *Yijing* and *Laozi*.

In the text of the *Yijing*, the term "柔" (softness) is often associated with "顺" (compliant), embodying a type of virtue. Just as the *Explaining the Trigrams* (说卦 shuogua) states, Kun (坤 pure Yin) means submissiveness and pliancy ([Lynn 1994](#), p. 122), because the image of Kun symbolizes the Earth, which yields to and is in harmony with the Heavens. The *Wenyan* 文言 also says, "The Dao of Kun is its compliance, embracing the will of Heaven and executing its tasks". Yu Fan's commentary on "Kun soft and yielding" ([Lynn 1994](#), p. 113) of *The Hexagrams in Irregular Order* (杂卦 zagua) states that Kun represents the yin,

which is harmonious and compliant, hence it is soft. Therefore, in Wang Bi's commentary on the *Yijing*, we often encounter the term "柔顺" (gentle and compliant) being used to represent the virtues of humility and obedience. However, in the hexagram Qian (乾), Wang Bi attempted to explain the respective drawbacks of the path of hardness and gentleness: "if one were to take up a position of headship over men by using nothing but hardness and strength, that would result in people not going along with it. If one were to engage in improper behavior by using softness and compliance, that would result in a Dao of obsequiousness and wickedness" (Lynn 1994, pp. 139–40). Thus, being compliant does not mean that people should blindly obey and submit, or even make reluctant concessions, because they should uphold the central and upright path (中正) as the baseline for their behavior. In the hexagram Kun (坤), Wang Bi continued to interpret the virtue of the Kun hexagram as compliance. However, when explaining that by sticking to compliance one can achieve beneficial outcomes, he treats "compliance" as a moral character trait. He was not satisfied with the traditional rationale in the *Yijing* that the earth supports the heavens, and pointed out that the logic behind this behavior is the relationship between the subject and the situation. He believed that the fortune of a hexagram largely depends on how well a person responds to the situation revealed in the hexagram (Hon 2010, p. 83). Taking the hexagram Da You (大有, Great Holdings) as an example, the Yin line is in the esteemed fifth position, being selfless towards objects, dealing with them with trust, and engaging with them with a compliant and upright attitude. The result is the realization of what Laozi described as teaching without words. Wang Bi referred to the virtue of the fifth line, which represents gentleness situated in the middle, as "信德" (trustworthiness or faithfulness in virtue). When it comes to the sixth line, where firmness rides over gentleness, Wang Bi termed it as "思顺" (thinking in compliance). He believed that in such a state, one can remain unburdened by material concerns and uphold noble aspirations. Although one might not fully grasp the essence of softness inherited in virtue, there is still something praiseworthy about it.

From the analysis above, we can conclude that Wang Bi emphasized different aspects of the meaning of "softness" (柔) in the texts of *Laozi* and *Yijing*. In his interpretation of the traditional virtue of compliance in the *Yijing* text, he pointed out on one hand the potential issues of extreme compliance, and on the other hand, he attempted to integrate Laozi's philosophy of "softness and pliancy" into the virtue of compliance, transforming the essence of compliance into Daoist principles like non-action, absence of desire, teaching without words, and non-interference with external matters. In Wang Bi's annotation of the third Yin of the hexagram Song (讼, Contention) and the second Yin of the hexagram Sui (随, Following) and Guan (观, Viewing), there are the phrases "体夫柔弱，以顺于上" and "体于（分）柔弱[5]" that discuss the nature of Yin. Richard John Lynn translated this as "The substance of Yin is soft and yielding" or "Yin is imbued with softness and weakness". For example, in the hexagram Sui 随, the second Yin is not independent; it must follow the one to which it is near (the first Yang) and abandon the interaction with the fifth Yang, because it could not maintain its proper goal. Therefore, this line is located at a time of following, which in substance is soft and pliant. However, this translation seems to overlook the verb implication of "体" (to embody or experience). In Wang Bi's explanation, when people act in a gentle and compliant manner, it is because they have grasped the essence of softness and pliancy. By Wang Bi's understanding, this means they have comprehended the essence of "nothingness". Through the function of this "nothingness", people adopt a stance of stillness and retreat in the tangible world. For the third line of the hexagram Song, which is positioned between two Yang lines and relates to conflict, it cannot act in the manner of the two Yang lines to "contest from below". In Lynn's translation, this is reflected as "Not being encroached upon, it safeguards all that it has" (Lynn 1994, p. 173). Regarding the attitude towards litigation, Wang Bi cited the arguments from *Analects* and *Laozi*, which are: "In listening to litigation [song], I am like other men. But what is really necessary is the prevention of litigation itself from happening!" (*Analects* 12, p. 13), and "Thus those who have virtue tend to their contracts and do not lay blame on others" (Chapter 79 in *Laozi*).

Therefore, it is believed that regardless of whether a dispute is resolved successfully, such success is temporary and will ultimately result in negative consequences. He held that the root of disputes lies in the division of things, therefore, the focus should not be on the litigation itself, but on preventing the potential for disputes from the start. For those in a weaker position when facing litigation, understanding conflicts from a non-litigious perspective allows them to handle disputes with gentleness and compliance. The second Yin in the hexagram Guan is situated in the lower trigram, which is why there is not a sufficiently broad view to observe things. "This is Viewing as through the crack of a door" (Lynn 1994, p. 262). Wang Bi believed that being in such a situation is like a sheltered woman, aware of her own ignorance, who can only deal with everything in a gentle and compliant manner. Zheng Kai 郑开 noted that under the influence of Confucian thought, various virtues (such as filial piety, kinship, gentleness, etc.) are subsumed under the umbrella of "benevolence and righteousness". However, in Daoist and Yin–Yang philosophical traditions, these concepts have been preserved and innovatively developed with meanings such as intrinsic nature and the law of nature (Zheng 2009, pp. 13–14). His understanding of "compliance" is in line with this. He reinterprets the compliant behavior driven by external forces as the result of an individual's proactive choice, transforming the Confucian virtues based on educational needs into a philosophy rooted in the intrinsic nature of softness and pliancy.

### 3. Individual Life and Social Management

There is a very close connection between Yin–Yang and hardness–softness, as both are derived from people's observations and summaries of the experiential world. However, when it comes to the position of hardness and softness, they have totally different views. Laozi advocated the great power of softness, and stated "The unyielding and mighty shall be brought low; The soft, supple, and delicate will be set above. 强大处下，柔弱处上" (Chap. 76, Mair, p. 150). However, according to *The Commentary on the Appended Phrases* (系辞 *Xici*), "The hard and the soft exist as hexagrams only after Yin and Yang have combined their virtues, for it is in this way that the numbers of Heaven and Earth become embodied in them and so perfectly realize their numinous, bright virtues. 阴阳合德，而刚柔有体，以体天地之撰，以通神明之德" (Lynn 1994, p. 86). Yin and Yang symbolize heaven and earth, representing the power of rising and falling, so Yin is below and Yang is above. This relationship between Yin–Yang and its position is called "carrying and riding" 承乘, which reflects the order of respect and humility between heaven and earth. In social management, it is the ethical norm of the king's respect and minister's humility. Among the six hexagrams, the odd numbers (first, third, and fifth) are taken as Yang, and the even numbers are taken as Yin. If the unbroken line is in the Yang position and the broken line is in the Yin position, it is in the right position 当位, otherwise it is out of position 不当位.

Although Wang Bi continued to accept the theory of "carrying and riding", he introduced the perspective of "time 时", trying to display the limitations of the principle of Yang's riding on Yin and bring the principle of softness from Laozi into the interpretation of *Yijing*. He pointed out that "the hexagrams deal with moments of time, and the lines are concerned with the states of change that are appropriate to those times 夫卦者，时也；爻者，适时之变也" (Lynn 1994, p. 29). The interpretation of hexagrams and lines should not be simply attributed to the judgment of good or bad, but should be judged comprehensively according to the specific circumstances and people's behavior under the situation. Wang Bi innovated traditional methods of interpreting the *Yijing*, summarizing new rules for interpreting the hexagrams and their lines. In particular, he presented new perspectives on the understanding of the Yin lines and their positions. First, in the *General Remarks on the Changes of the Zhou*, Wang Bi pointed out that neither *The Commentary on the Images* nor *The Commentary on the Appended Phrases* (系辞 *Xici*) contains the principles of right position and out of position (determining the position of the first line and the sixth line by Yin and Yang). The question of divination was only mentioned in the discussion of the upper lines of the hexagram Qian and Xu. Wang Bi concluded that there is no positioning problem in the first and sixth line because they refer to the beginning and end state of things. His in-



terpretation breaks the traditional correspondence between Yin–Yang and their respective positions, denying the authoritative status of Yang over Yin. He also separated them from the relationship with fortune and misfortune, creating an interpretive space for the Yin in odd positions within the hexagram, thereby elevating the actual status of Yin in the *Yijing*. Second, Wang Bi proposed the perspective of emphasizing the minority ("以寡为宗"). He elaborated on the explanation of line positions from the *Tuan Zhuan* (彖传) (Zhu 1995, p. 257). He believed that when interpreting the meaning of a hexagram, there was "the body or substance of a hexagram" (一卦之体). This means he perceived the hexagram as a whole; as long as one grasps the concept of the core line, this provides an understanding of how to act in complex situations. In his interpretations, Wang Bi specifically pointed out that in hexagrams with five Yang lines and one Yin line, the solitary Yin line represents the main subject of that hexagram. This can be observed in hexagrams such as Guai (夬), Gou (姤), Tong Ren (同人), Lü (履), Xiao Xu (小畜), and Da You (大有).

Furthermore, in his interpretations of the hexagrams, Wang Bi emphasized the virtue of softness, treating it as a person's ability for self-restraint and self-discipline. Regarding his interpretation of the hexagram Lin (临, Overseeing), this hexagram is combined with Dui 兑 (below) and Kun 坤 (above). When compared with hexagram Kun 坤 and Fu 复, it presents the image of the Yang (the hard) gradually growing stronger. Wang Bi described that as the Yang cycle progressively waxes, the Dao of Yin daily wanes; with the growth of hardness, the softness in one's being is put in a dangerous state. Wang emphasized the right position of the fourth Ying; because of the obedient Ying, it does not dread the growth of hardness and strength, moreover, its response to the first Yang helps to build a harmonious interaction between these two. Wang Bi argued that this hexagram symbolized the Dao of the noble man who increases day by day, and the Dao of the petty man who increasingly comes to grief day by day (Lynn 1994, p. 254). It manages to avoid blame because softness has its own virtue and does not violate what is right. Softness and hardness, or Yin and Yang, are the core elements of *Yijing*, revealing the laws of the development of things through their changes. In contrast, Laozi emphasized the mutual transformation of the two but particularly highlighted the leading role of softness. In Wang Bi's commentaries, while he still stressed the positive role of hardness within the empirical world, he emphasized the role of softness in helping the subject improve on a transcendental level. According to He Shao's 何劭 biography of Wang Bi,

> Pei Hui 裴徽 held the position of a Court Gentleman at the Ministry of Personnel at that time, and Wang Bi paid a visit to Pei before reaching the age of maturity. Pei knew immediately that this was an extraordinary person. Therefore, he asked, "Non-being is, in truth, what the myriad things depend on for existence, yet Confucius was unwilling to talk about it, while Laozi expounded upon it endlessly. Why is that?" Wang Bi replied, "Confucius experienced the form of Non-being, which cannot be explained in words, and that is why he did not talk about it. But Laozi, by contrast, was an advocate for Being, and thus always discussed the deficiency of Non-being". (Lou 1999, p. 645)

In his debate with He Yan 何晏, Wang Bi proposed that the sage should be emotionless and engage with objects without being entangled by them. That is, even when confronting matters with a robust and proactive attitude, one should be able to transcend them, using the humility, emptiness, and subservience advocated by Laozi to dissolve the troubles that ensnare through fame, favor, humiliation, and gain or loss. Using the concepts of root and branch, as well as essence and function, he reinterpreted the pairs of firmness and gentleness, and Yin and Yang, with gentleness as the root and appreciated the virtue of being soft in different circumstances, especially in difficult times.

Just like Laozi, Wang Bi's emphasis on softness was not limited to the cultivation of personal virtue. He advocated for maintaining softness not only as a moral requirement for rulers but also as an effective strategy for state governance and conflict resolution. Wang Bi put forth the argument of "处尊以柔" (being gentle in a noble position). In the structure of the six lines of *Yijing*, the middle positions, namely the second and fifth lines, are held

in esteem. Among them, the fifth line is considered the most prestigious, leading to the ancient description of rulers as having the dignity of "九五之尊" (the honor of the fifth Yang). Wang Bi accepted the important role of the fifth Yang. In addition, he also regarded the fifth Yin as having a central role in understanding the hexagram. As mentioned above, in the Da You (大有) hexagram, where there is only one Yin line, the fifth Yin with its yielding nature occupies a noble position. All lines above and below will respond to and welcome it. In the hexagram Jin (晋, advance), the fifth Yin is a bright ruler; he does not flaunt his intelligence. Instead, he orderly delegates tasks to his subordinates. Hence, in Wang Bi's annotation of Chapter 49 of the *Daodejing*, he stated:

> [Thus] things will have their principle, and affairs will have their master [without anything being distorted]. Once that is the case, it is possible for [me, the Sage Ruler,] to let the pearl strings of [my] mian hat obscure [my] eyes without fear of being deceived; to let the yellow pillows stuff [my] ears without concern about slanderous comments. Furthermore, what is the purpose of [the ruler's] exerting the intelligence of his single body to spy out the sentiments of the Hundred Families? 物有其宗，事有其主。如此，则虽冕旒充目而不惧于欺，璜纩塞耳而无戚于慢。又何为劳一身之聪明，以察百姓之情哉! (Wagner 2003, p. 283)

This manner of ruling, where one delegates to subordinates without showing off one's capabilities, is what Wang Bi referred to as the technique of being "soft and yielding". Here, Wang Bi replaced the Confucian approach of governing and educating with benevolence, righteousness, rituals, and rules with the Daoist approach of a ruler who governs by non-action. He believed the ideal monarch should treat the state and his subjects with a mindset that is gentle and receptive, embodying the Daoist principles over the traditional Confucian emphasis on governance. In his annotation of Chapter 58, he says:

> [A ruler] who is good at regulating government will have neither shape nor name, neither [government] activity nor standard that could be pointed out. [His government] is "hidden [from view]" [but] eventually will bring about the Great Order. That is why [the text] says: "He [a ruler] whose government is hidden [from view]". His people will have nothing to struggle about and compete for, wide and grand [they are in their] generosity; that is why [the text] says: "will have his people be generous". (Wagner 2003, p. 318)

> 言善治政者，无形、无名、无事、无政可举。闷闷然，卒至于大治。故曰"其政闷闷"也。其民无所争竞，宽大淳淳，故曰"其民淳淳"也。

Wang Bi pointed out that those skilled in handling governmental affairs are not merely rulers for governance, but rather rulers in pursuit of the Dao, who are dedicated to returning to a state of non-action. He attempted to integrate the Confucian concept of delegation with the Daoist idea of non-action by using the principle of softness. In his annotation of the fifth Yin in the hexagram Lin (临, overseeing), he pointed out that:

> Fifth Yin is situated in the noble position, treading there in such a way that it manages to practice the Mean. It knows how to receive the hard and strong [Second Yang] with decorum and thereby strengthen its practice of rectitude. Fifth Yin does not dread the growth of Second Yang's strength and so is able to employ Second Yang in its service. It is by employing others in order to extend one's abilities, while doing no wrong in the process, that the perspicacious can extend his powers of sight and hearing to the utmost and the one empowered with wisdom can fulfill his ability to plan. This is how such a one accomplishes things without purposeful effort and reaches goals without having to take the steps himself. (Lynn 1994, p. 257)

> 处于尊位，履得其中。能纳刚以礼，用建其正，不忌刚长而能任之。委物以能而不犯焉，则聪明者竭其视听，知力者尽其谋能，不为而成, 不行而至矣!

A similar expression can be found in his annotation of the fifth Yin in the hexagram Ding (鼎, The Cauldron):

Fifth Yin abides in centrality with its softness and weakness and thereby is capable of thoroughly implementing the principles of things. This one is the beneficiary of the strength and correctness [of Second Yang], so the text says: "[The Caldron has] yellow ears and metal lifters, so it is fitting to practice constancy". As the ears are yellow, it is able to receive what is strong and correct in order to have itself lifted up. (Lynn 1994, p. 456)

居中以柔，能以通理，纳乎刚正，故曰"黄耳金鉉，利貞"也。耳黄，则能纳刚正以自举也。

Wang Bi believed that softness has a strong inclusivity. It can resonate with and encompass the firm and robust second Yang (九二) in this hexagram. By delegating to capable ministers, it allows subordinates to manage the country diligently according to their duties and abilities. Hence, Wang Bi particularly emphasized that rulers possessing the virtue of gentleness should treat their officials with humility, putting themselves in a lower position and acting with sincerity and trustworthiness. On one hand, this approach circumvents the practical difficulties of Laozi's philosophy of governance through non-action. On the other hand, it incorporates the Confucian principle of governance through rituals and laws. Furthermore, it provides a metaphysical explanation and practical supplement to the philosophy of the Huang–Lao School, which advocates for the ruler practicing non-action while the ministers take action. Therefore, Wang Bi commented on the hexagram *Weiji* (未济, Ferrying Incomplete):

Fifth Yin occupies the exalted position with softness and weakness and is located at the height of civility and enlightenment, for this is the ruler of the Ferrying Incomplete hexagram. Thus it is that this one must behave with rectitude, for only then should he have good fortune. And only with such good fortune should he manage to avoid regret. Thanks to his character, which consists of softness and compliance, civility and enlightenment, this one occupies the exalted position in such a way that he entrusts responsibility to the capable and does not attempt to take charge of everything himself. He tempers military action with his civil virtues and modifies hardness and strength with his tenderness, all of which truly represents "the glory of the noble man". Such a one entrusts responsibility to another because of his ability and does not harbor suspicions about him. Thus that other [Fourth Yang] exerts himself to the utmost and achieves meritorious success in the conquest [over the Demon Territory, i.e., the world's troubles]. Thus the text says: "The sincerity he has… brings good fortune". (Lynn 1994, p. 549)

以柔居尊，处文明之盛，为未济之主，故必正然后乃吉，吉乃得(无)悔也。夫以柔顺文明之质，居於尊位，付与于能，而不自役，使武以文，御刚以柔，斯诚君子之光也。付物以能，而不疑也，物则竭力，功斯克矣，故曰"有孚，吉"。

"Wenming" (文明), according to Kong Yinda 孔颖达's annotation of the hexagram Qian, means "articulate and bright". Wang Bi believed that a monarch, by governing the nation with the virtues of softness and compliance, can integrate both the civil (文) and martial (武) approaches, mastering a balanced approach of firmness and gentleness, to achieve the nation's prosperity and strength.

## 4. Conclusions

In the context of Yin and Yang, softness is the result of individual cognition in the contrast between firmness (刚) and gentleness. However, regarding the outcome of this cognition, Laozi referred to this self-awareness of contentment and knowing when to stop as "clarity" (明). Within the Daoist perspective, softness and pliancy are neither objects of salvation nor opportunities to arouse sympathy; they are manifestations of the Dao's nature in the real world. Wang Bi's annotations differentiated between actions of softness with compliance and a consciousness of softness and pliancy, which takes the intention of external behavior into account. This intentionality encompasses ignorance, absence of desire, and stillness, and it reflects humanity's grasp on the essence of "nothingness".

Using the philosophical structure of root (本) and branch (末), Wang Bi interpreted gentle behavior as a natural expression following the grasp of the essence of softness and pliancy. Compared to annotations from scholars before him, Wang Bi greatly amplified the practical value of softness. Softness not only becomes an effective method for individuals to preserve themselves but also an essential way to continuously acquire life energy. More importantly, he rationalized Laozi's contrarian thought, making it an ideal means for rulers and social governance. In addition to considering compliance as a moral value, he emphasized viewing softness as a natural characteristic of life, a cognitive approach for humans to understand and perceive the Dao.

**Funding:** This research was funded by the National Social Science Fund of China, grant number 19BZX052.

**Conflicts of Interest:** The author declares no conflict of interest.

## Notes

[1] In Chapter *No Duality* (不二) it said, "Laodan esteemed softness, Confucius benevolence, Mo Di Wholess…". (17/7.1), John Knoblock and Jeffrey Riegel trans. *The Annals of Lü Buwei: A Complete Translation and Study*, Stanford: Stanford University Press, 2000, p. 433.

[2] The Chapter *Yiwen Zhi* (艺文志) of *Hanshu* commented on Laozi as "He protects himself with purity and emptiness, and sustains himself with humility and pliancy". and the General Narration to Daoist Category of *Siku Quanshu* said Daoism "using softness to overcome hardness, and seeing retreat as advancement".

[3] Mair, Wagner, and Lynn translated Ruo 弱 as "weakness". however in this paper, it is translated as "pliancy" (for a specific interpretation, please refer to the second part of this paper). The translations referenced from Wagner and Lynn in this paper will still use "weakness".

[4] Han Fei said, Goujian entered service in Wu and personally took up arms to wash horses for the King of Wu. This allowed him to later kill Fuchai in Gusu (Suzhou). When King Wen was berated at the king's gate, his complexion did not change, and King Wu captured King Zhou of Shang in the Muye battlefield. Thus it is said: "keeping soft is stronger". 勾践入宦于吴, 身执干戈为吴王洗马, 故能杀夫差于姑苏。文王见詈于王门, 颜色不变, 而武王擒纣于牧野。故曰："守柔曰强。"

[5] In *Annotations and Explanations of Wang Bi's Collected Works* 王弼集校释 by Lou Yulie (楼宇烈), a distinction is made between the annotations of the characters "于" and "分" in the Hexagram Sui (随). It is argued that the character "于" is not in alignment with the intended meaning of the annotation, whereas the character "分" is deemed appropriate, meaning '本分one's proper role.' This interpretation is also recorded in Ruan Yuan 阮元's *Zhouyi Jiaokan Ji* (周易校勘記), where it is noted that in this context, "分" is used in several different versions. (Lou 1999, p. 306).

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
