# Peer review of "The Practice of Rou 柔 from Wang Bi’s Perspective"

_religions, doi:10.3390/rel14121470_

Round 1

Reviewer 1 Report

Comments and Suggestions for Authors

Comments on the Quality of English Language

editing suggestions

2, 90-91  fan as "negative opposite"--reversion?

4, 155-54 explain "rule by facing south"

8, 351 "telegram"?

9, 391 "supper"?

10, 448 "violet"?

10, 487 "mian"?

check for a few other issues with spelling and grammar

Author Response

The article has undergone English language editing by MDPI. The spelling and grammar problem has been checked.Please see the attachment.

Reviewer 2 Report

Comments and Suggestions for Authors

The article compares the different views of Laozi and Wang Bi on "softness" which elucidates the different philosophical emphasis in the text and in Wang Bi’s commentary. The first section of the paper discusses the Laozi’s view on softness, followed by an examination of the views on softness found in the Zhuangzi, Han Feizi, and Yan Zun. On one hand this offers an interesting historical review, yet it doesn’t seem to have a strong and logical connection on the discussion on Wang Bi in the second part of the paper. I suggest the author clarifies the connection.

In the conclusion, the author argues that Wang Bi “refuses to view softness as a moral value. Instead, he sees it as a natural characteristic of life, a cognitive approach for humans to understand and perceive the Dao. The question if the Laozi considers softness as “a moral value” remains open.  It would be helpful to clarify of softness as a moral value to begin with.

1.     This paper has many typos and mistakes that need to be fixed. For example:

a. Page1, line 21, “shuld” should be “should”.

b. The font and the size of the characters should be identical.

c. Some book titles are prefixed with "the" and some are not.

d. Some quotes have blank lines before and after them while others do not.

2.     The language needs to be improved as well. For example:

a.     Page1, line 3 and 35, “the thesis” should be “the paper” or “the article”.

b.     The tenses of the paper are very confusing and need to be revised carefully.

c.     å©´å„¿ is translated as “baby” or “babies” in some places and as “infant” in other places, the translation should be identical.

Comments on the Quality of English Language

The language demands revisions, preferably by a native english speaker. 

Author Response

The grammar and mistakes mentioned in the review has been modiffied and this paper has undergone English language editing by MDPI. The question of whether Laozi considers softness as “a moral value” is open, and Wang Bi sitll cosidered compliance as a moral value. Also I have tried to build the connetctions between the commentaries before Wang Bi with Wang Bi.

Reviewer 3 Report

Comments and Suggestions for Authors

It is hard to find research findings and significance in this article. 

1. After reading the whole text, it fails to deliver a clear research problem statement and purposes. Why so important to study Wangbi's Rou? What is the main point or something new the author wants to show us in this study? How does this study engage with current scholarship? The article lacks a literature review too. 

2. In sections 1 and 2, the comparison of the terms Rou is too lengthy and lacks coherence. The author should interpret Laozi's Rou concept under its Taoist philosophy context, rather than compare it with different schools. Is this comparison important in understanding Wangbi's thought? By the way, Laozi's Rou is not necessarily equivalent to weakness.

3.   The format is inaccurate and makes the text untidy. Some of the terms should add Chinese words. For example, Guan Yin, Lao Dan, Yijing, Daodejing...   

The citations should change footnotes to notes behind the main text.

Comments on the Quality of English Language

The author should smoothen the language and make all the sentences clear. 

Author Response

The manucript has beed revised with the language problems with the help of MDPI. Some of the terms has added Chinese words.

Round 2

Reviewer 3 Report

Comments and Suggestions for Authors

Still, the author should elaborate more on the introduction section I have mentioned. More specifically, the literature review, the research's problem statement, and its research contributions in this field. 

Also, I can't find the author's response to my previous comments. The author response file submitted by the author was not related to the feedback.   

Author Response

Thanks for the suggestions. I have modified the paragraghs on the commentaries before Wang Bi in Part I, trying to demonstrate the commentaries before Wang Bi failed to provide detailed guidance on how to understand the function and how to apply the methods of softness.
